# The Biocompatibility of Wireless Power Charging System on Human Neural Cells

Henrieta Skovierova [1],*, Miroslav Pavelek [2], Terezia Okajcekova [3], Janka Palesova [3], Jan Strnadel [1], Pavol Spanik [2], Erika Halašová [1] and Michal Frivaldsky [2],*

[1] Biomedical Center Martin, Jessenius Faculty of Medicine in Martin, Comenius University in Bratislava, Malá Hora 4C, 036 01 Martin, Slovakia; jan.strnadel@uniba.sk (J.S.); erika.halasova@uniba.sk (E.H.)

[2] Department of Mechatronics and Electronics, Faculty of Electrical Engineering and Information Technologies, University of Zilina, 010 26 Zilina, Slovakia; miroslav.pavelek@fel.uniza.sk (M.P.); pavol.spanik@feit.uniza.sk (P.S.)

[3] Department of Medical Biochemistry, Jessenius Faculty of Medicine in Martin, Comenius University in Bratislava, Malá Hora 4D, 036 01 Martin, Slovakia; terezia.okajcekova@uniba.sk (T.O.); janka.palesova@uniba.sk (J.P.)

\* Correspondence: henrieta.skovierova@uniba.sk (H.S.); michal.frivaldsky@feit.uniza.sk (M.F.)

**Abstract:** The progress in technology and science leads to the invention and use of many electrical devices in the daily lives of humans. In addition to that, people have been easily exposed to increased newly generated artificial electromagnetic waves. Exponential use of modern electronic devices has automatically led to increase in electromagnetic wave exposure. Therefore, we constructed the prototype of wireless power charging system to study the biocompatibility of electromagnetic field (EMF) generated by this system on various human cell lines. There are many studies indicating the negative bio-effect of EMF on various types of cells, such as induction of apoptosis. From the other point of view, these effects could rather be beneficial in the way, that they could eliminate the progress of various diseases or disorders. For that reason, we compared the impact of EMF (87 kHz, 0.3–1.2 mT, 30 min) on human normal as well as cancer cell lines based on morphological and cellular level. Our results suggested that EMF generated by wireless power charging systems does not have any detrimental effect on cell morphology, viability and cytoskeletal structures of human neural cells.

**Keywords:** wireless power transfer; electromagnetic field; standardization; cytotoxicity assay; neural cell lines; cell morphology

## 1. Introduction

Technological growth in human life, especially in the development of electrical and communication technologies has resulted in increased exposure to artificial electromagnetic fields (EMF). People use the newly developed wireless technologies in cell phones or computers daily. As a result, living organisms are being exposed to artificial EMF that they have not experienced before. The effect of radiofrequency-EMF (RF-EMF) on living/biological systems has been controversial due to many studies with various results. Moreover, the International Agency for Research on Cancer (IARC), which is a part of World Health Organization (WHO), was assigned to define the link between use of mobile phones and head/neck cancers. Therefore, IARC has classified RF-EMF as possibly carcinogenic to humans (Group 2B) [1].

Many people, especially young people, use the wireless connection every day. The possibilities of exposure to considerable doses of EMF waves is all around us. Therefore, the social interest of the effect of RF-EMF exposure has been increased [2]. The research on RF-EMF effect on human health is very controversial. Many studies have been focused on negative impact of RF-EMF on human health and development of cancer [3], neurological [4,5] and reproductive disorders [6,7], immune dysfunction [8–10], genetic damages [11], cognitive effects [12], and electromagnetic hypersensitivity [13]. Moreover,

the detailed knowledge regarding the mechanism of the effects of RF-EMF on biological processes has not been elucidated clearly. Some recent studies highlighted that RF-EMF produced from mobile phones is absorbed by the brain in level, which can affect its activity [4,14]. Additionally, the increased thermal effect of RF-EMF on neural cells could lead to changes in some physiological functions of neuronal cells [15,16].

Electromagnetic waves are generated using various electronic devices. Living systems can absorb produced waves. These waves cause vibration of polar or charged molecules, which are components of plasma membrane or cytoplasm of cells. Therefore, RF-EMF could be critical to human health and safety. Based on this, specific absorption rate (SAR) standards were specified by International Commission on Non-Ionizing Radiation Protection (ICNIRP). SAR refers to the amount of ratio wave energy absorbed by unit mass of human body (W/kg). ICNIRP has adopted a conservative position and uses 4 W/kg averaged over 30 min as the RF-EMF exposure level [17,18]. To ensure that thermal effects are avoided, safety factors have been incorporated into exposure limits, resulting in whole-body-averaged limits of SAR to 0.08 and 4 W/kg in uncontrolled and controlled environments, respectively [17].

As the wireless charging becomes an increasingly discussed issue addressed not only by academic researchers, but also by industrial subjects, standardization is required for a smooth and reliable commercialization. It should allow interoperability between different chargers and increase safety. Therefore, the standards must include safety and efficiency criteria, EMF limits, and interoperability targets along with wireless charging testing. Among the most important standards for WPT is SAE TIR J2954/2 [19], which is concerned with alignment methods, interoperability, and frequency band and power levels. It establishes the efficiency greater than 90% when using matched coils and higher than 85% when using interoperable systems. Since the wireless charging is accompanied by an intermediate to high-frequency EMF, it is also necessary to instruct the EMF levels. Another guideline is established in IEEE Standard for safety levels with respect to human exposure to RF-EMF, 3 kHz to 300 GHz [20].

Several studies focused on the possible impact of electromagnetic waves generated by EMF on neurons have recently been published with great interest, especially in relation to brain tumor development [1]. In addition, the increased risk observed in some of the epidemiological studies is inconsistent with the stable frequency of occurrence of oncological diseases in the population. There are discussable results according to experimental conditions, models of study or used methods. The effect of IM-MF was studied on the process of development to analyze the differentiation processes during embryogenesis. It was proved that 21 kHz MF does not have any toxic effect on animal embryonic cell lines and their differentiation [21]. However, a further study found frequency-dependent effects on cell proliferation but without a clear trend [22].

Recently, only few in vitro studies examined the effect of WPT used for charging of electrical vehicles on human health [23]. The aim of present study is to investigate the impact of an electromagnetic wave (magnetic field intensity, 0.3–1.2 mT and frequency 87 kHz) on various human neural cell lines. Furthermore, we compared exposed and control cells on various levels, such as (i) cell morphology and adherence using light microscopy; (ii) cell viability using spectrophotometry; (iii) ratio of living, apoptotic and dead cells determined by flow cytometry; (iv) compactness of the cell cytoskeleton analyzed by fluorescent microscopy. These analyses could point to the safe use of newly generated prototypes of power wireless charging systems on human brain cells, which could lead toward neurodegeneration or tumor development.

## 2. Material and Methods

### 2.1. Design of Experimental Thermo Incubator

The specific experimental incubator was required due to maintenance of the constant temperature within the EMF exposure for in vitro cultivated human neural cells. The geo-

metrical design of proposed incubator is shown on Figure 1, while it is seen that it considers two chamber design (outer vacuum chamber and inner-temperature controlled chamber).

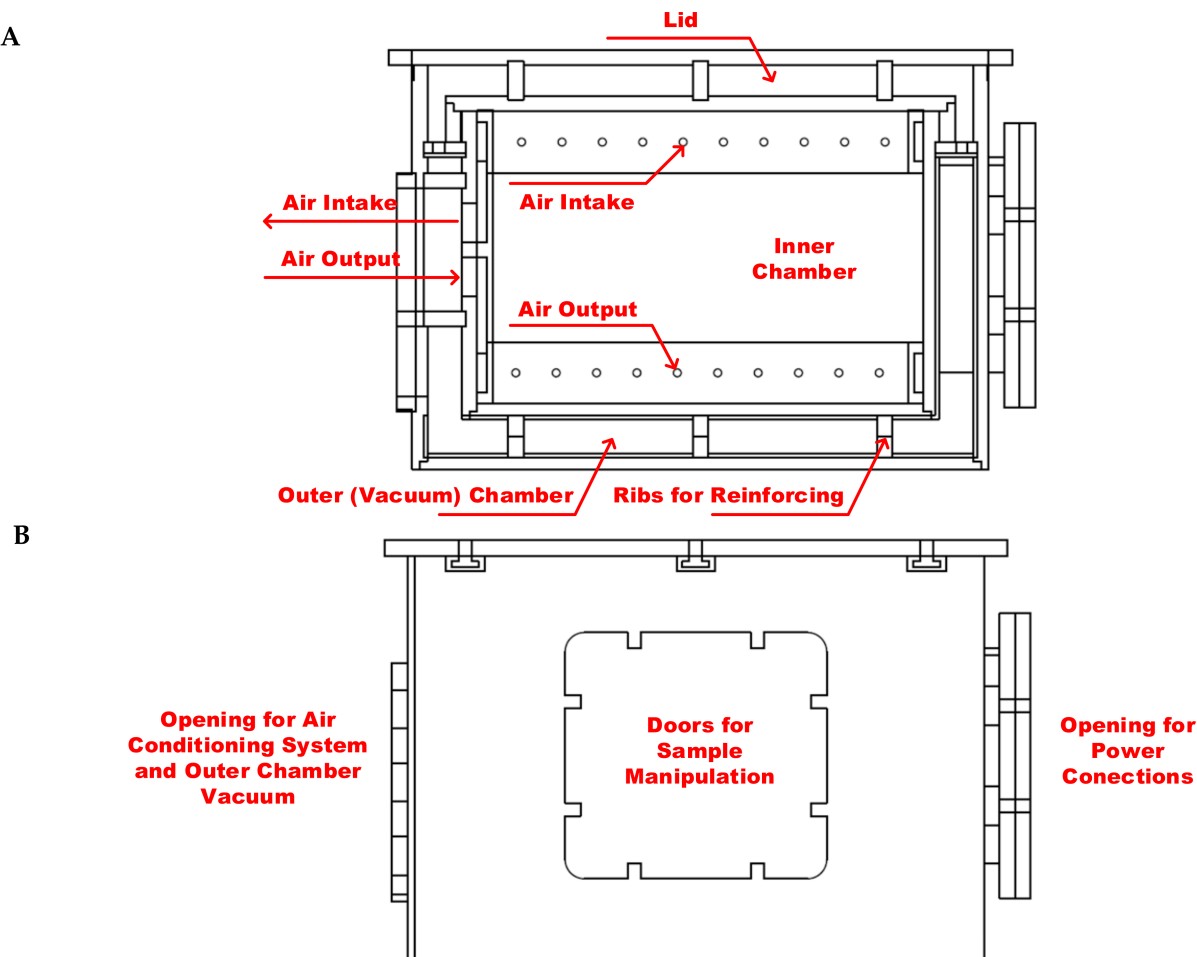

**Figure 1.** The scheme of thermo incubator (**A**)—top view, and its side view (**B**).

It is required that temperature during the tests is close to human temperature body. Therefore, the heating system was implemented within the incubator, while regulation of the temperature was within 35.5–37 °C. Material of the system is 15 mm acrylic glass, whereby the outer chamber is reinforced by ribs with holes to outstand the vacuum. The incubator has three air sealed apertures (air inlet/outlet and vacuum for outer chamber, opening for power connections for WPT system and opening for biological samples manipulation). All three apertures are fully reconfigurable as they are independent parts of the incubator system. The inner chamber of the incubator has outlet air openings around the upper part of the chamber and inlet air openings around the lower part of the chamber. The incubator has the option to open the upper part "Lid" to allow the insertion of the WPT system (this part is also air sealed). All air sealed parts are connected to the incubator using 3D printed screws. In the future, it is expected also to add $CO_2$ level control within incubator environment.

Figure 2 is showing that the air heating is done in an external chamber for air conditioning (50 cm from inner chamber of incubator to ensure that the air-conditioning system as well as the measurement of the WPT influence on the samples are not affected). The airflow within the air conditioning system is made by the 40 mm server at the inlet side of the system. The air temperature is adapted by switching high current through the power resistors at the outlet of the air conditioning system and then it is monitored in small compartment at the end of the air-conditioning system. The temperature regulation

algorithm was working in hysteretic mode to ensure that the temperature was within the defined range, while µC Arduino Nano was used as main control system.

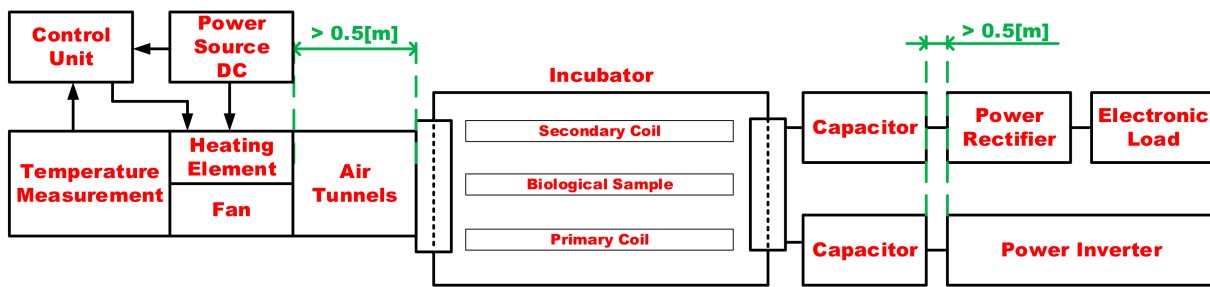

**Figure 2.** Block diagram of WPT measurement within the incubator with the air-conditioning system.

## 2.2. WPT System Operational Settings and Scaling of the EMF for Exposure

The configuration of the WPT system for the experimental purposes of the EMF exposure is shown in Figure 3. It comprises a laboratory power supply unit, primary side high-frequency inverter which supplies primary side transmitting coil with compensation capacitor. The secondary side (receiving side) is composed of a receiver, compensation capacitor, high-frequency rectifier, and programable electronic load. The intensity of the EMF was modified through the PSU input voltage of the WPT system, while the identification of the EMF intensities was identified through the use of finite element method (FEM) model [24].

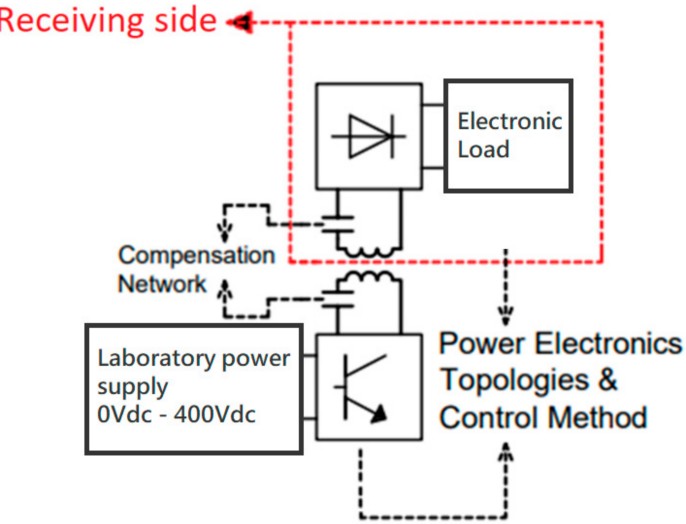

**Figure 3.** Block diagram of the configuration of the WPT system.

The operational settings of the proposed WPT system reflect the use of neural human cells [25]. The system of the coupling coils was designed as a reconfigurable system considering mutual power transfer distance and electromagnetic shielding. The coil's matrices were placed on the construction which can easily change mutual distance of the coils within the range of 0.05–0.25 m (Figure 4). As was described earlier, the magnetic intensity for the selected exposing component should be at the maximum level of 1.2 mT at the frequency of EMF 87 kHz. To identify the distribution of the EMF around the designed coupling coil system, FEM analysis was provided to identify Input/Output parameters of the WPT system. The analysis was reconfigurable according to the use of a shielded, or nonshielded coupling coil system. The use of the shielding is important to reduce negative impacts of radiated EMF above the system of the coils. The proposed shielding

system also enables investigation of the various geometrical arrangements on the EMF distribution [24–26].

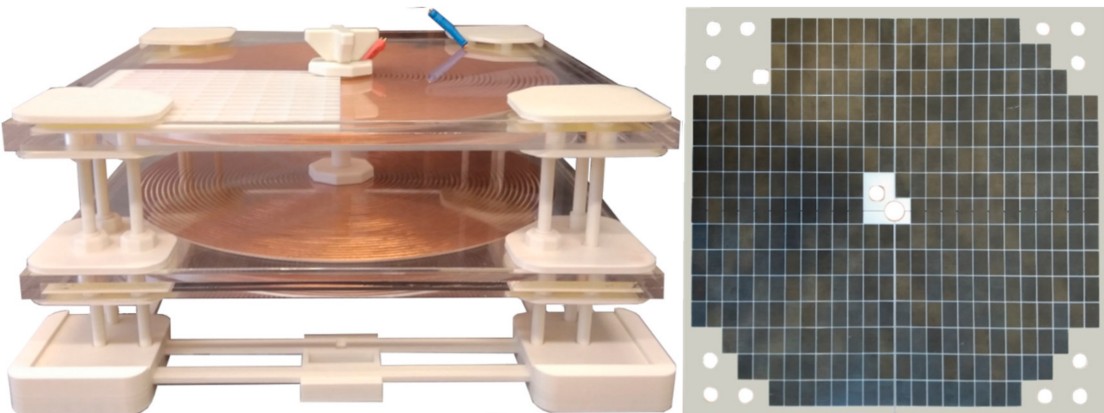

**Figure 4.** System of coupling coils with adjustable transfer distance (**left**) and reconfigurable magnetic shielding (**right**).

### 2.3. Simulation Analysis of the EMF Intensity and Distribution within Proposed WPT System

The need for the identification of the magnitudes of magnetic field is required before experiments with EM exposure will be realized. There are several ways to identify EMF intensities, while the first one can be done experimentally and the second through the use of a verified simulation model. As the laboratory instrumentation for experimental measurement of EMF radiation could be complex and expensive, we utilized the second possibility, i.e., highly accurate simulation model of proposed WPT system [24].

The equivalent circuit used for the identification of the EMF intensities is shown in Figure 5. The model considers the electrical and magnetic domain, while the electrical circuit settings are given in Table 1. Figure 6 represents arrangement and location of the measuring points within the set-up.

**Table 1.** Settings of the electronic circuit simulation model.

| Circuit Element | Value | Point (+) | Point (−) |
| --- | --- | --- | --- |
| Ground | GND | 0 | 0 |
| Voltage source | VMAX*sin($\omega$t) | 0 | 1 |
| Capacitor 1 | $1/(\omega\char94 2*\text{comp1.mf.LCoil\_1})$ [F] | 1 | 2 |
| Resistor 1 | RCoil1 [$\Omega$] | 2 | 3 |
| External I vs. V1 | Coil voltage (mf3/coil1) | 3 | 0 |
| External I vs. V2 | Coil voltage (mf3/coil2) | 0 | 6 |
| Resistor 2 | RCoil2 [$\Omega$] | 6 | 7 |
| Capacitor 2 | $1/(\omega\char94 2*\text{comp2.mf2.LCoil\_1})$ [F] | 7 | 8 |
| Load | RLOAD [$\Omega$] | 8 | 0 |

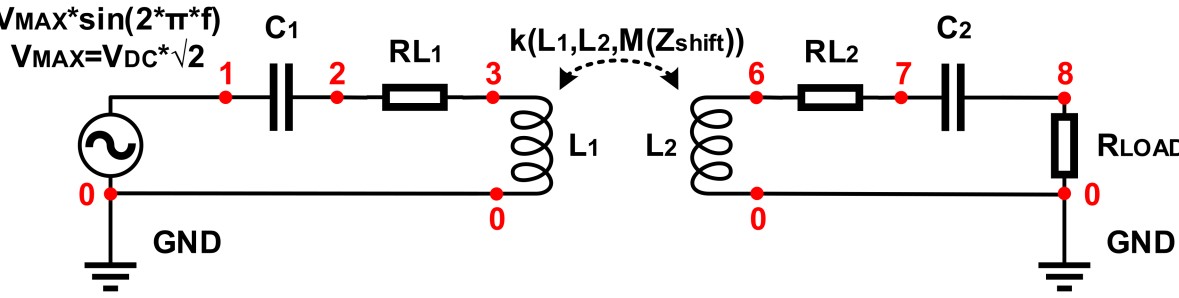

**Figure 5.** Electrical circuit definition within the electrical domain of the FEM simulation model.

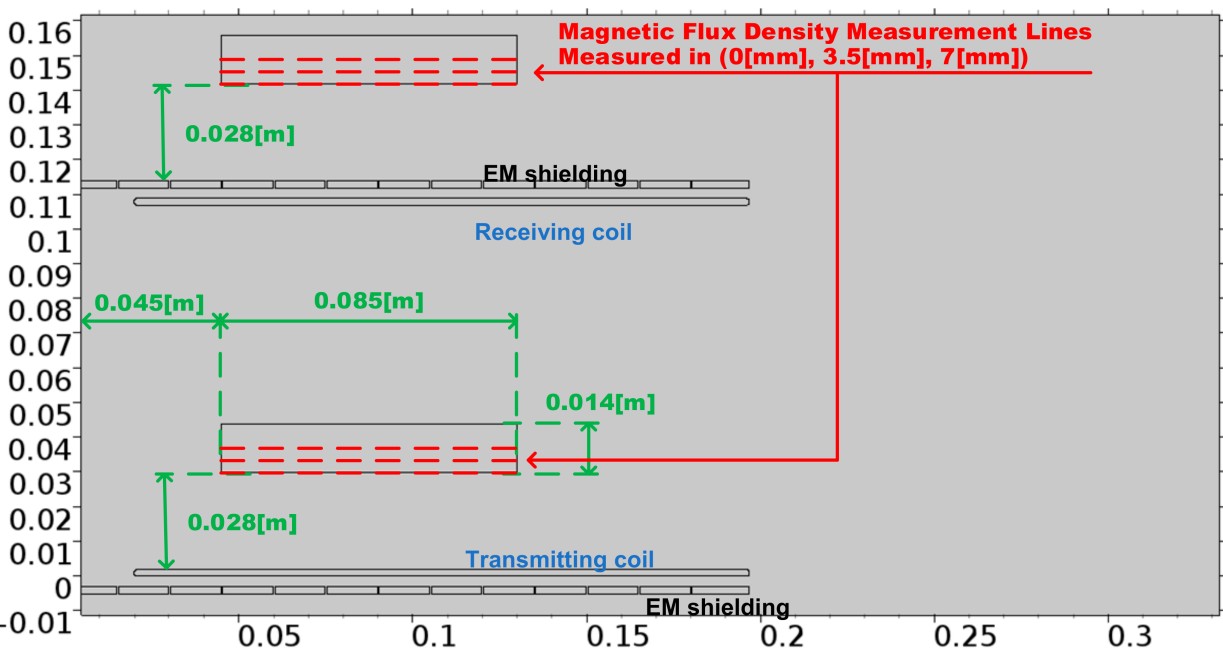

**Figure 6.** The scheme of the biological samples location in the WPT system during exposure.

Magnetic flux was identified within the area, where neural cells (sample) were located, while three cutting lines per sample were evaluated. It was defined that samples were located within the area between the transmitter and receiver (28 mm from transmitter), and within the area above the receiver (28 mm from the secondary coil).

To identify magnetic field intensities, which are required for the exposure of neural cells, the parametric simulation analyses were performed to identify the target value of magnetic induction. The example of the simulation result is shown in Figure 7. As can be seen in Figure 7, approximately 1.2 mT was achieved on the bottom exposure sample, and approximately 0.3 mT was exposed on the top sample, while Input/Output parameters of WPT system for experiments are identified as follows: Input DC voltage 249.5 V; Resistive load 48.9 Ω; Main Resonant Frequency 85 kHz—switching frequency of high-frequency inverter; Higher Resonant Frequency 87 kHz.

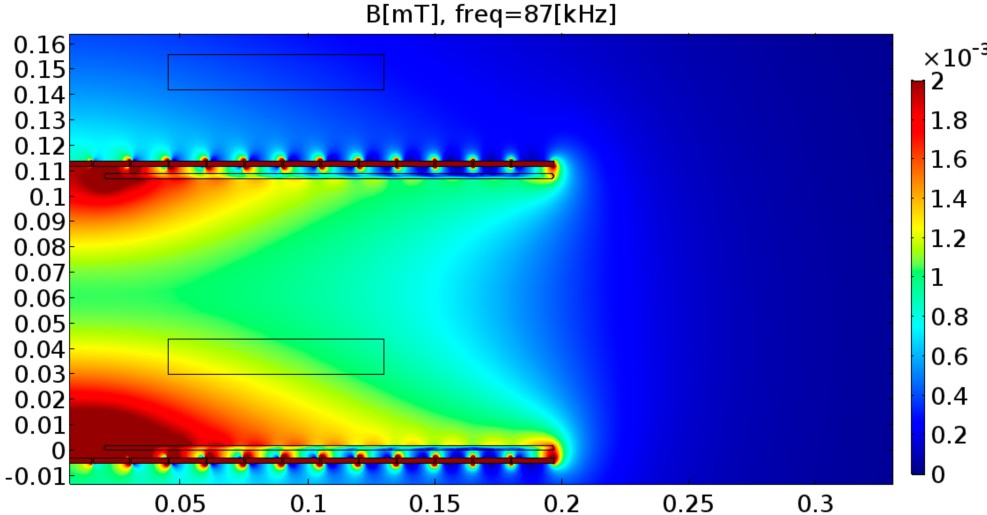

**Figure 7.** Simulation results of the distribution of magnetic induction within a system of coils (operating frequency, 87 kHz; transmitting power, 1450 W).

Investigating the frequency characteristic for the above parameters, Figure 8 represents dependency of the transmitted power on the switching frequency of the inverter. The value of power transfer for the investigated EMF exposure is approximately 1.5 kW.

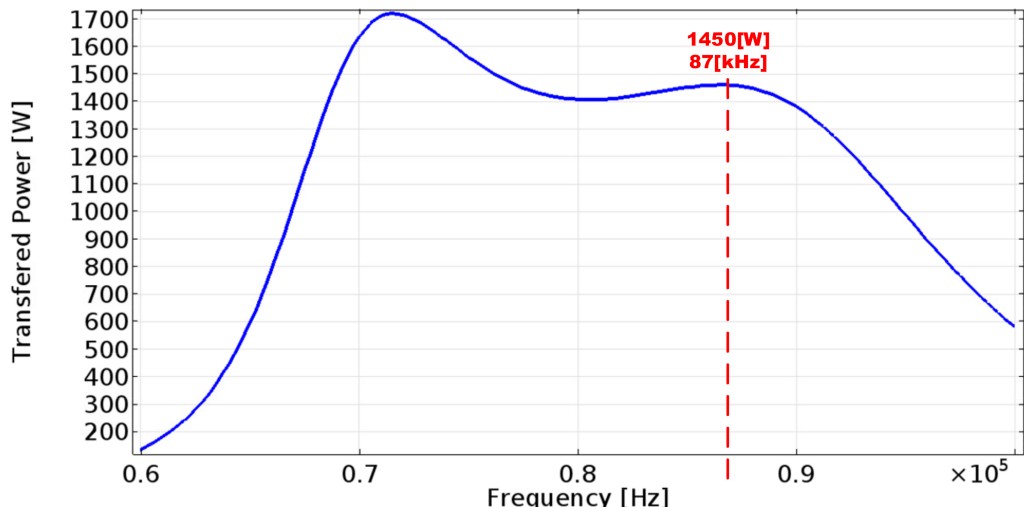

**Figure 8.** Frequency characteristic of the WPT system for the EMF distribution from Figure 6.

It must be noted that the above parameters of the WPT system represent scaled exposure because of the use of neural human cells. If different exposure samples are considered, i.e., human tissues, or larger biological structures, the EMF intensities should be higher and can be modified in the way of variation of Input/Output parameters of the WPT system.

### 2.4. Cell Cultures and Cultivation Conditions

Three commercially available human neural cell lines were used in the present study. Human astrocytes HA (ScienCell, Carlsbad, CA, USA) were grown in astrocyte medium (AM, ScienCell) supplemented by 2% fetal bovine serum (FBS, ScienCell), $1\times$ astrocyte growth supplements (AGS, ScienCell) and 100 I.U./mL of penicillin and 0.1 mg/mL of streptomycin (P/S, ScienCell). The neuroblastoma cell line (SH-SY5Y, ECACC, Salisbury, UK) was grown in Dulbecco's modified Eagle's medium/Ham's nutrient mixture F12 (DMEM/F12 = 1:1, Sigma-Aldrich, Darmstadt, Germany) supplemented by 10% fetal bovine serum (FBS, Sigma Aldrich) and 100 I.U./mL of penicillin and 0.1 mg/mL of streptomycin (P/S, Biosera). Glioblastoma cell line (T98G, ECACC) was grown in DMEM high glucose (Sigma-Aldrich) supplemented by 10% fetal bovine serum (FBS, Sigma-Aldrich) and 100 I.U./mL of penicillin and 0.1 mg/mL of streptomycin (P/S, Biosera, Nuaille, France). Cells were cultivated at standard conditions (5% $CO_2$, 37 °C, humidified atmosphere). HA and T98G were plated at the density of $5 \times 10^3$ cells/cm$^2$, SH-SY5Y was plated at the density of $1 \times 10^4$ cells/cm$^2$. Cell morphology was analyzed by light microscope under an inverted phase contrast (Optika XDS-2, Ponteranica, Italy), magnification $100\times$.

### 2.5. Exposure of Cell Cultures

Before exposure, cells were grown up to 30–40% confluence at standard cultivation conditions. Then, cells were divided into four groups, (i) control (unexposed) group; (ii) exposed and located on coil, (iii) exposed and located between coils, and (iv) exposed and placed on shielding. The cells were exposed to one-time continuous radiation of 87 kHz EMF for 30 min at 0.3–1.2 mT in the preheated cell incubator of 35.5–37 °C. The cells in culture flask/microtiter plate were located as in mentioned in Figures 6 and 7. The control group of each cell line, unexposed cells, were during this 30 min period cultivated at 37 °C out of any artificial exposure of EMF. These parameters were set up based on the range limits of the prototype. After the exposure period, cells were cultivated for another 44 h at

standard conditions prior to further analysis. This prolonged cultivation assesses the late effects of EMF exposure on cells. Moreover, during this period at least one new generation of cells is obtained and then, a potential damaged effect of EMF could be more noticeable.

### 2.6. MTT Cell Viability Assay

To analyze the possible cytotoxic impact of EMF on cell viability, MTT assay was performed. It is a simple biochemical analysis based on enzymatic reduction of yellow MTT to purple formazan [27]. Briefly, cells were seeded in 96-well microtiter plates as 12 replicates for each cell line. After the time of regeneration (44 h), cells were rinsed by DPBS (Biosera) and further incubated in fresh cultivation medium containing MTT (Sigma-Aldrich) for another 5 h at 37 °C in the humidified atmosphere. Then, 5% SDS (*w/v*; Sigma Aldrich) was added to dissolve formazan salt. After 18 h of incubation, colorful changes were measured spectrophotometrically at 540 nm using Synergy™ H4 Microplate Reader (BioTek, Bad Friedrichshall, Germany). The relative viability of cells was determined as the ratio of O.D. value of formazan produced by treated cells to O.D. value of formazan produced in control group and expressed as the percentage of control. The O.D. of control cells was considered as 100%. All experiments were performed at least three times. The results of MTT assay were expressed as mean ± standard deviation (SD) of 12 replicates as indicated. Statistical analysis was performed using one-way analysis of variance (ANOVA) and significance of difference between means with $p < 0.05$ considered statistically significant.

### 2.7. Annexin V Assay and Flow Cytometry Analysis

To identify apoptotic and necrotic cells after EMF exposure, Annexin V conjugated with FITC fluorescent label was used. By binding of Annexin V to phosphatidylserine (PS), which is strictly located in the inner part of the plasma membrane, apoptotic cells could be identified. To help to distinguish between live, necrotic and apoptotic cells, 7-amino-actinomycin D (7-AAD) dye was used. 7-AAD is excluded by live and intact cells. Therefore, the ratio of live, early apoptotic, late apoptotic, and dead cells could be determined. Briefly, cells after 44 h of regeneration were passed through a cell strainer (Corning, Durham, NC, USA) to prepare a single cell suspension. Then, cells were resuspended in Annexin V Binding buffer (BioLegend®, San Diego, CA, USA) and incubated with Annexin V dye followed by addition of 7-AAD viability staining solution for 15 min in the dark. After addition of Annexin V Binding buffer, labeled cells were analyzed by flow cytometry (FACS Aria™ Cell Sorter, BD Bioscience, San Jose, CA, USA).

### 2.8. Immunocytochemistry

Using specific antibodies, Phalloidin, which selectively binds to actin filaments of cellular cytoskeleton the cell motility, cell division, cytokinesis, and organelle movement as well as cell shape could be analyzed. Neural cells after exposure and regeneration were washed by DPBS and fixed by 4% Paraformaldehyde (Cell Signaling Technologies, Danvers, MA, USA) for 30 min at room temperature. After washing, cells were incubated with Alexa Fluor®488 Phalloidin (Cell Signaling Technologies; 1:1000 in solution of 1% BSA in DPBS) for 80 min followed by another labeling by DAPI (Sigma Aldrich) for 10 min at room temperature in dark condition. Cells were washed three times by DPBS and analyzed by fluorescent microscope (WiScan®, Hermes IDEA Bio-Medical, Rehovot, Israel), magnification 100×.

## 3. Results

Based on the simulation analyses, we adjusted operational parameters (Input/Output) of the physical sample according to previously received results. The coupling coils system were placed within a thermo incubator and the biological samples (as was mentioned in Figures 6 and 7) were placed between the coils, as well as above the receiving coil (Figure 9). The operational frequency of WPT system as well as the frequency of EMF

was equal to 87 kHz, transmitted power was 1.5 kW and the value of magnetic induction within individual cells varied from 0.3 to 1.2 mT based on the location. Compared to the standards issued by ICNIRP 2010 (the limit is 27 μT), the experimental settings exceed the limit in range 11–44 times. These increased values of induction were set to test more extreme conditions, compared to the limit ones. We compared the effect of EMF exposure on normal (astrocytes, HA) as well as on cancer (neural SH-SY5Y and glial T98G) cells.

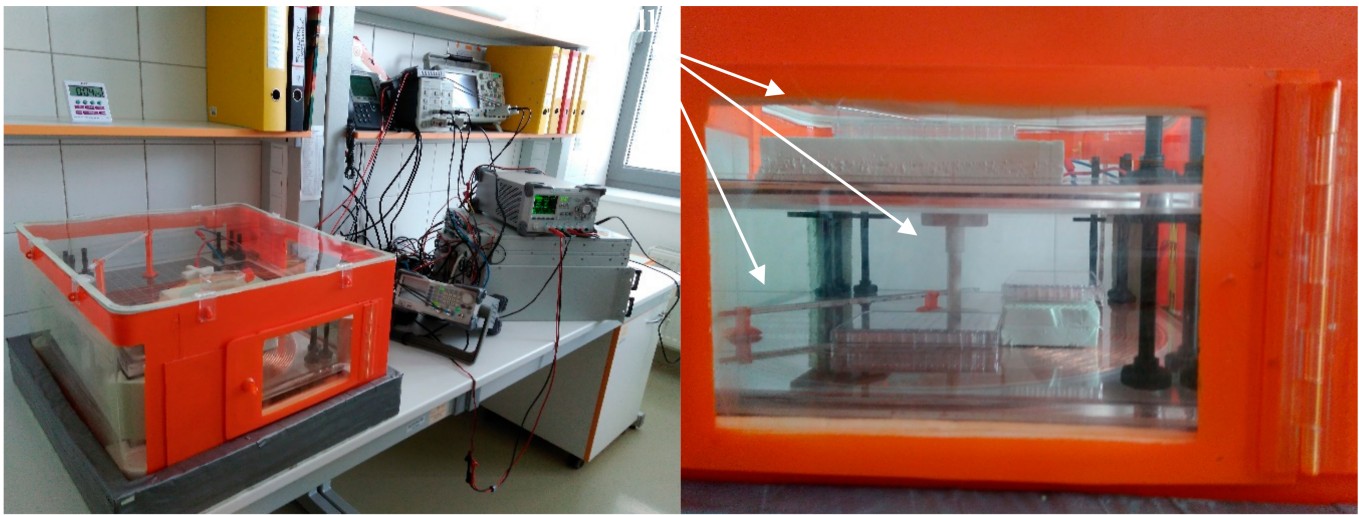

**Figure 9.** Experimental set-up of the wireless power transfer system (**left**) and detail of location of biological samples within the coils coupling system (**right**).

### 3.1. The Effect of EMF on Cell Morphology and Viability

To evaluate changes in cell morphology and density, we analyzed neural cells by light microscopy immediately after EMF exposure as well as after 44 h of regeneration (Figure 10). We did not observe any changes in cell morphology or affection of cell adherence, neither any dead cells presence.

Normal healthy astrocytes (HA) retained their elongated shape, the cells were mitotically active even after 44 h of regeneration ($2.1 \times 10^5$ cells in control; $2.0 \times 10^5$ cells in exposed sample). Similar observations were found in samples with cancer cells. T98G and SH-SY5Y did not have their proliferative activity negatively affected after 44 h of regeneration compared to the control group of cells ($5.5 \times 10^5$ cells in control; $5.3 \times 10^5$ cells in exposed sample).

Subsequently, we analyzed the impact of EMF on cell viability and basal energy metabolism using the MTT assay (Figure 11). In normal astrocytes (HA), we observed a slightly increased metabolic activity up to 12%, depending to location in EMF. Moreover, there were no significant differences in cell proliferation between exposed and control samples. Cancer cells (T98G as well as SH-SY5Y) were exposed along healthy cells. Therefore, we could compare the obtained results between different cell types. There was an increase in cell viability in cancer cells (6–10% in T98G, 2–9% in SH-SY5Y), but the increase was not statistically significant. These results may suggest that exposure of neural cells to 87 kHz, 1.2 mT for 30 min could not trigger any cytotoxic responses and is likely not harmful to cells.

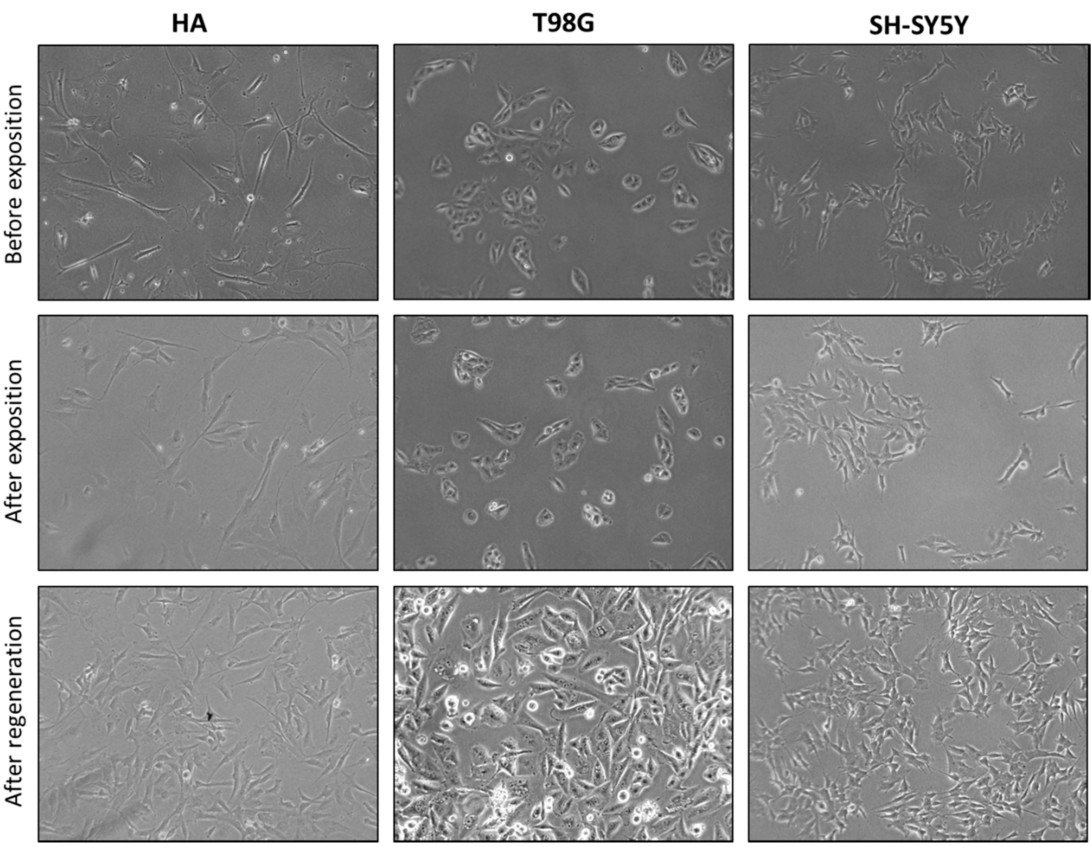

**Figure 10.** Cell morphology of human neural cells (HA, T98G, SH-SY5Y) analyzed by light microscopy. Cells before EMF exposure (first row), cells directly after exposure (87 kHz EMF for 30 min; second row) and cells after 44 h of regeneration in standard conditions (third row). Representative photomicrographs were taken at phase contrast at magnification 100×.

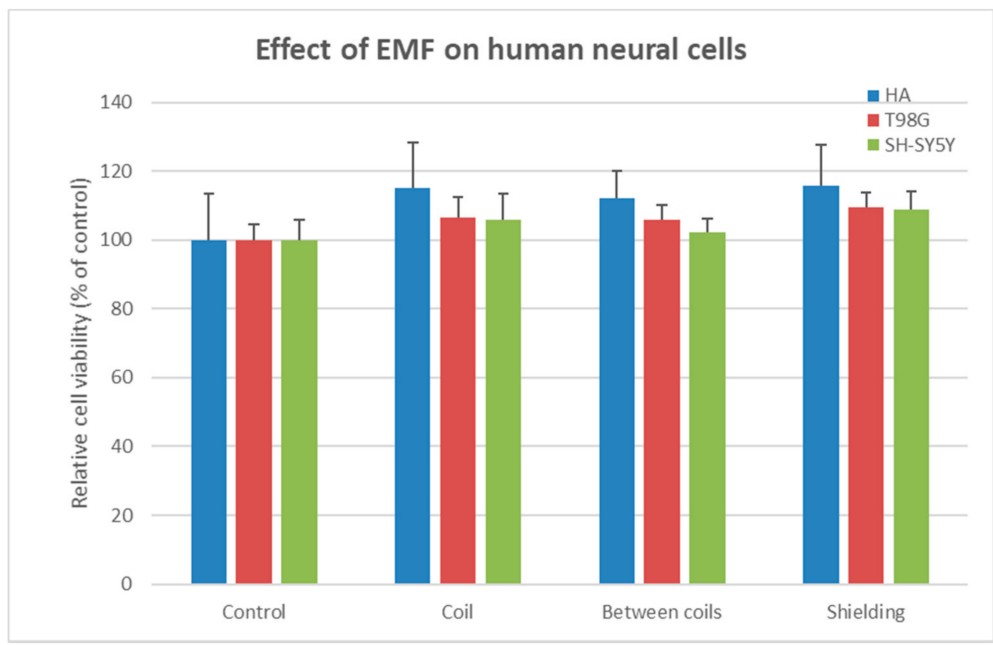

**Figure 11.** Effect of EMF on cell viability of HA, T98G and SH-SY5Y analyzed by MTT assay. Cells were exposed to 87 kHz for 30 min and then regenerated for another 44 h at standard cultivation conditions. The control group represents 100% viability. Data are mean ± SD. Data are representative of three independent experiments with 12 replicates each.

## 3.2. The Effect of EMF on Cell Death and Inner Cellular Organization

To analyze the ratio between live and apoptotic cells, flow cytometry analysis was used. We compared control group with exposed groups located as mentioned in Figure 6. The observed results are summarized in Table 2.

**Table 2.** The effect of EMF (87 kHz, 30 min) on neural cell death analyzed by flow cytometry. After exposure, cells were regenerated for another 44 h and analyzed by Annexin assay.

|  | Location | Living Cells | Early Apoptotic | Late Apoptotic | Dead Cells |
|---|---|---|---|---|---|
| **HA** | Control | 87.3% | 4.7% | 6.2% | 1.8% |
|  | Coil | 88.5% | 4.6% | 4.5% | 2.3% |
|  | Shielded | 89.9% | 3.5% | 3.5% | 3.0% |
| **T98G** | Control | 86.2% | 4.1% | 6.5% | 3.2% |
|  | Coil | 86.1% | 4.0% | 6.5% | 3.3% |
|  | Shielded | 87.3% | 4.0% | 5.7% | 3.0% |
| **SH-SY5Y** | Control | 84.9% | 4.3% | 7.1% | 3.7% |
|  | Coil | 83.7% | 4.8% | 8.5% | 3.0% |
|  | Shielded | 87.8% | 3.0% | 5.1% | 4.1% |

Considering normal HA cells, we found that the highest percentage of viable cells (89.9%) was presented for the group located on the receiving coil equipped by magnetic shielding. In contrast, the control group had the lowest percentage of viable cells (87.3%). However, the number of dead cells was the lowest considering the control group (1.9%) compared to the group located on shielded receiver (3.0%). However, these differences are minimal and such variability is accepted from a biological point of view. In addition, cells that are metabolically and mitotically active also have a certain minimum percentage of dead or apoptotic cells. We obtained very similar values in the case of cancer cells (T98G, SH-SY5Y). The percentage distribution of cells in followed processes (apoptosis, necrosis or viable cells) is comparable and independent of type of cell line. Overall, SH-SY5Y cells showed slightly lower values at percentage of viable cells, which may be due to their biological nature, or the fact that after 44 h of regeneration, the cells were slightly overgrown and some of them could naturally activate cell death. Based on the results it could be concluded that the group of dead cells represents the lowest percentage within each group, thus supporting the fact that the test conditions did not result in increased death of cells.

We evaluated the effect of EMF on cytoskeletal organization of neural cells (87 kHz, 1.5 kW, 30 min) and followed by 44 h of regeneration under standard conditions (Figure 12). The control group of healthy HA cells had normal structures that exhibit sharply rounded nuclei with noncondensed chromatin. The impact of EMF did not result in the visualization of fragmentation of nuclei into small apoptotic bodies, as evidence of the negative effect of radiation on cells. The actin fibers of the cytoskeleton of the cells were dense, distinct and organized in parallel bundles. Moreover, the structure of the cytoskeleton is very compact and markedly limited, no loosened or disassembled fibers were observed.

The cancer cells (T98G, SH-SY5Y) after exposure had preserved similar inner cellular structures compared to control groups. We did not observe any changes in cytoskeletal structures and organization, which is location independent within the WPT system, neither between normal and cancer cells.

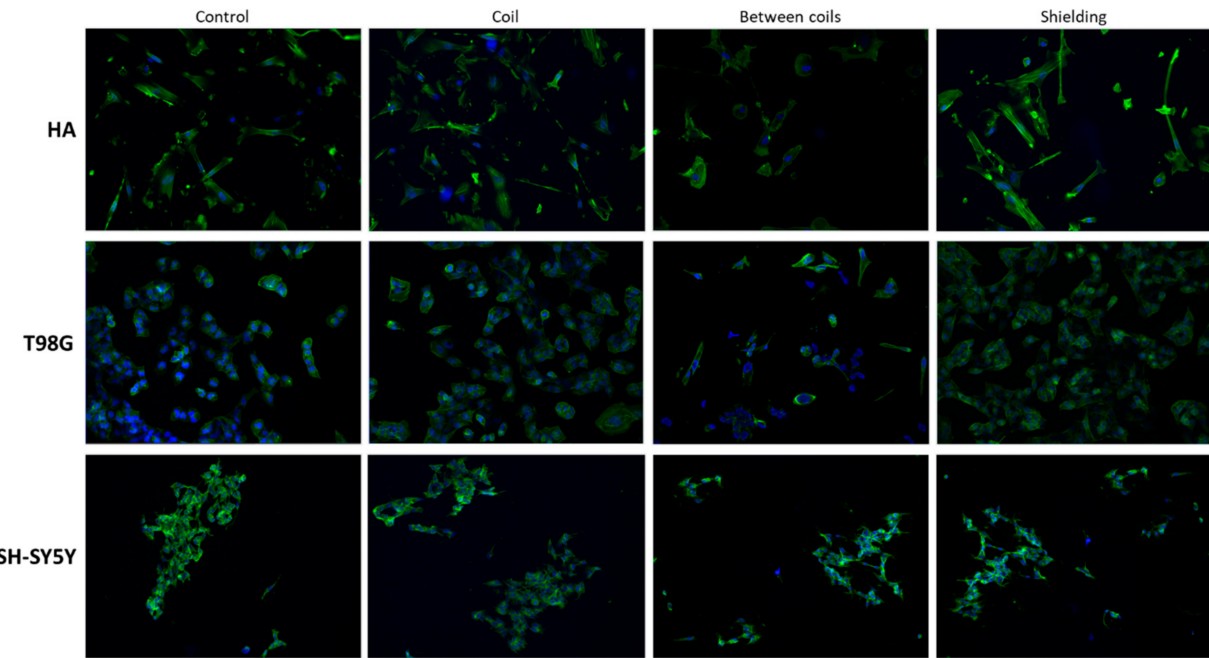

**Figure 12.** Fluorescent analysis of human neural cells (HA, T98G, SH-SY5Y) after exposure to EMF (87 kHz, 1.5 kW for 30 min) followed by 44 h regeneration. Cytoskeleton of cells was visualized by Phalloidin dye (green) and nuclei were stained by DAPI (blue). Representative photomicrographs were taken by WiScan® Hermes System at magnification 100×.

## 4. Discussion

The use of various electrical devices in daily human life has grown rapidly. It is due to a technological progress in the last two decades. Wireless communication technologies, such as smartphones, have become a necessity for modern people. Nowadays, EMF generated by mobile cell phones, radio, satellites communication systems, Wi-Fi systems or TV increase the level of electromagnetic smog noticeably. Each person is potentially exposed to various sources of EMF at the same time. The International Agency for Research on Cancer (IARC) classifies the RF-EMF as Group 2B, it means possibly carcinogenic to human [28]. Moreover, it has been hypothesized that neurological problems could occur as a consequence of RF-EMF exposure due to the small distance of the cranial nervous system and the location where the smartphones are predominantly used. These neurological disorders include changes in sleep habits [29], headache [30], and changes in EEG [31,32]. However, there are several studies indicating more harmful nonthermal effects on human health [33–38].

The aim of our study was to analyze the biological effect of EMF, which is generated by a newly constructed prototype for a wireless power charging system. Neural cells are one of the most frequently exposed cells using the contactless charging system. In addition to that, neural cells are very sensitive to environmental pollution ("electromagnetic smog"), because they participate in the transmission of excitation within the nervous system and their regenerative ability is low. We used three different cell lines. One of them is healthy normal cells, the other two are cancer cell lines. We used the biological variety of cells to get more complex results about the effect of EMF on cellular level. We also investigated if our set of parameters for EMF generation could have a selective inhibition influence on cell growth of cancer cells while normal cells would be not affected. Human Astrocytes (HA) were used as a model to study the function of the central nervous system and interaction between neural cells. Astrocytes are glial cells found in the brain and spinal cord. They are responsible for maintaining, supporting and repairing nervous tissue. In laboratory, HA are used as model to study neurotoxicity, drug development and various neurological diseases (Parkinson's or Alzheimer's diseases) [39,40]. The neuronal properties of SH-SY5Y, human neuroblastoma cells, makes these types of cells as a valuable model to study various

pathological processes and mechanisms at molecular and morphological level [41]. The last one, T98G cells are derived from human glioblastoma multiform tumors. This cell line is used as a model for testing drug cytotoxicity and new drug development, which can be applied for brain cancer research [42,43].

WHO defined the range from 300 Hz to 1MHz as IF-EMF as a nonionizing radiation [44]. We applied 87 kHz during all experiments, which is defined within allowed operational range by relevant normative ICNIRP 2010. The Amplitude/Intensity of exposure field was in the range of 0.3–1.2 mT and exposure time was 30 min. These experimental values were set based on the maximal range limits of the constructed prototype along with the evident proof of the impact of EMF generated by the WPT system on human cells. The magnetic flux density, which was used in present work, is approximately 11–44 times higher comparing to reference level (27 µT), which is recommended by ICNIRP guidelines [17].

We analyzed cell morphology, viability and cytoskeletal integrity in three cell lines in exposed cells comparing to unexposed control cells. After exposure, cells were regenerated for another 44 h at standard cultivation condition to see the possible cumulative negative impact of EMF on cells. Based on our set conditions, we were not able to detect any diverse effect on cell morphology, viability, or cytoskeletal disintegration at 87 kHz, 1.2 mT for 30 min. Several studies have analyzed electric, magnetic and EMF exposure effect, obtaining the most diverse outcomes [1,15]. Most of the studies were performed using cell or animal models. They have provided basic information on the possible biological effects of EMF exposure to living systems. There are few studies focused on the acute or chronic EMF exposure in IF range and its consequence on human health at cellular level in vitro [45–47]. Therefore, the investigation of potential adverse or cumulative EMF effect is quite challenging and precise epidemiological studies are needed to confirm the possible negative effects of EMF exposure to humans [15]. Moreover, the technology of the WTP system (85 kHz) was studied on human body models [48]. They evaluated the induced electric field in the human body models for a magnetic field leaked from a WPT system in an electrical vehicle. The induced electric fields in three anatomically based human body models were weaker than the basic restriction, although the external magnetic field exceeded the reference level. Even though these observations are supporting, it is necessary to verify their impact on living systems.

During our analysis, we did not observe any changes in cell morphology as well as in number of cells (Figure 10). The tested conditions did not have any negative impact on cell division on healthy normal cells, HA. We even did not find any dead cells accumulated in cancer cell lines and did not prove the apoptosis activated after exposure. Some papers observed altered cell metabolism or cell adhesion associated with changes in cell morphology [49,50]. Even though, the effect on calcium efflux and cell differentiation were also proved [51,52]. On the other hand, several in vitro studies did not find any significant effect of EMF exposure on cell differentiation, neither on phagocytosis or chemotaxis [53,54].

Alteration of proliferation is a sensitive indication that could be used to identify any cytotoxic impact. The effect of EMF on cell viability was analyzed spectrophotometrically by MTT test (Figure 11). Considering our results, cell viability after exposure to 87 kHz at 0.3–1.2 mT of EMF for 30 min did not appear to have an adverse effect on HA, SH-SY5Y or T98G cell lines. HA showed a little increase in cell viability, however, these changes were not statistically significant. Some in vitro studies were focused on the effect of MF on cell proliferation and viability. However, the outcomes were different, as these studies detected reduced or increased cell growth and viability, as well [23,41,45].

It has been shown that external EMF can induce a variety of molecular and cellular responses, including microfilament reorganization, changes in calcium dynamics, neuronal growth cone guidance or enhanced stem cell differentiation [55]. Even though electrotherapy has been successfully used for nerve fiber repair, bone fracture treatment or cancer chemotherapy, little information is known about the effect of IF-EMF on the cell mechanical properties [56]. The cytoskeleton is one of the most significant cellular mechanical

structures that allow elasticity and stability of the cell undergoing multiple deformations without losing its integrity. In addition, the important role of the cytoskeleton has been established in complex intracellular signaling pathways. It mediates cell responses, as changes in cell adhesion or in gene expression and secretion of extracellular matrix by itself structural rearrangement alterations [57]. Our results did not show any changes in actin filaments organization in normal human astrocytes. Even though, there was not any detrimental effect of 87 kHz EMF on cytoskeletal structures of cancer cells. This observation is in conformity with [58]. Most experiments were focused on the effect of EMF on tubular structures [59,60]. However, the results were discussable, mostly depending on EMF. Low frequency of EMF caused actin filaments reorganization from continuous, aligned structure to discontinuous globular patches. On the other hand, cells exposed to a higher range of EMF were not visibly affected [60].

Many studies oriented on MF or EMF exposure observed the most diverse outcomes, only a negligible amount of them consider acute or chronic EMF exposure in intermediated frequency and its influence on human health at cellular level in vitro [45]. We proved that IF-EMF at 87 kHz is well tolerated and without relevant response on cellular metabolism or morphology in various types of human neural cells. To get a more complex view about the effect of wireless charging power systems on human tissue, other molecular processes such as activity of oxidative stress enzymes, oxygen and nitrogen radical production as well as cell cycle should be analyzed.

From the technical point of view, other aspects need to be clarified and studied on human cells, e.g., the differences either in the effect or in the magnitude of the effect, in respect to the use of continuous and intermittent EMF. Innovative WPT technologies could find a place in various areas of human life. There is high potential to be applied in medicine. Boutry et al. constructed a pressure sensor, which was made from biodegradable materials to measure the arterial blood flow [61]. The technology, which worked wirelessly, could be used in post-operative monitoring of blood flow after reconstructive surgery.

## 5. Conclusions

So far, only a few in vitro studies have been focused on the impact of wireless power transfer systems within charging of the electric vehicles on human health. Therefore, we recently analyzed the effect of electromagnetic radiation of a WPT system prototype on cell morphology, viability and internal structures of human neural cells and highlighted the biosafety of using a de novo-generated prototype. On the other hand, if we could identify the conditions under which EMF has a possible negative effect on cancer cells, whether it is the induction of apoptosis or DNA disruption, as well as the breakdown and dysfunction of essential proteins, then it could be used as a supportive treatment during oncological treatment.

Based on our preliminary results, we can assume that the EMF generated by the optimized WPT systems under the tested conditions does not have a negative effect on the metabolic, mitotic and regenerative activity of human neural cells. In order to optimize and develop the system, further studies are needed to confirm the potential negative effects of radiation on living systems. Therefore, it is necessary to perform another set of experiments, which could conclude pulse and longer exposure, and more complex molecular analyses focused on protein expression during oxidative stress or compactness of DNA molecules.

**Author Contributions:** Research design, M.F., H.S.; measurement and experiments, J.S., T.O., J.P., H.S., M.P.; development, analysis of data, presentation of results H.S., M.F., P.S., E.H.; manuscript writing, M.F., H.S.; manuscript editing E.H., P.S. All authors have read and agreed to the published version of the manuscript.

**Funding:** This research received no external funding.

**Institutional Review Board Statement:** Not applicable.

**Informed Consent Statement:** Not applicable.

**Data Availability Statement:** Not applicable.

**Acknowledgments:** This work was supported by Grant APVV-17-0345 and "Operational Program Integrated Infrastructure 2014–2020 of the project: Innovative Solutions for Propulsion, Power and Safety Components of Transport Vehicles, code ITMS 313011V334, cofinanced by the European Regional Development Fund".

**Conflicts of Interest:** The authors declare no conflict of interest.

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
