# Peer review of "The Biocompatibility of Wireless Power Charging System on Human Neural Cells"

_applsci, doi:10.3390/app11083611_

Round 1
Reviewer 1 Report
- What is the novelty of the electrical part solution?
- Specify the description of the receiving part of the WTP system (Fig. 2 and Fig. 3).
Author Response
The response for reviewer is attatched as separate file.

Reviewer 2 Report
The manuscript written by Skovierova et al. is interesting and important. The manuscript is generally well written and seems to be appropriately conducted. Three different cell types were used in the study, which is very positive. However, I have some comments and questions to the authors:
- The authors used 87 kHz exposure in the study, and they define it as RF-EMF. Should it be rather intermediate frequency magnetic field?
- It would be nice to see the temperature data during the exposure, however the authors state that it stayed between 35.5-37 degrees and it was monitored.
- 5. Exposition of Cell Cultures. I think it should be exposure of cell cultures not exposition.
- In the same paragraph the authors write: Then, cells were divided into four groups, (i) control (unexposed) group; (ii) exposed and located on coil, (iii) exposed and located between coils and (iv) exposed and placed on shielding.
- Was there a sham exposure? If not, why?
- Were the controls located in incubator or where?
- How the authors chose the exposure time and level?
- In my opinion the 30 min exposure time was quite short and there could have been more timepoints.
- Later in the same paragraph the authors write: After exposition period, cells were cultivated for another 44 hours at standard conditions prior further analysis. This prolonged cultivation asses the late effects of EMF exposition on cells. Moreover, during this period at least one new generation of cells is obtained and then, a potential damaged effect of EMF could be more noticeable.
- It is also good to assess the delayed effects but why the effects were not measured immediately after exposure? Normally the immediate effects are tested first and then it is moved to delayed effects and possibly to genomic instability.
- In the beginning of the manuscript the authors mark the references with numbers but in the discussion both numbers and giving the reference with name and year are used. This is frustrating and the authors should be consistent.
- In the page 14, two references have been overwritten, please check if those should be there or not and also check the reference list.
Author Response

(The authors gave the same response as above.)

Reviewer 3 Report
The topic of the paper is really interesting, the experimental setup too,
however the frequency used is not in line with the type of devices you mentioned in the introduction and discussion.
The major issues are:
- In the introduction, if you define the acronym for artificial electromagnetic fields (EMF), please in the following lines use the EMF acronym to refer to the artificial electromagnetic fields;
- In the introduction, please define the acronym of WHO agency;
- In the introduction at the beginning you missed to tell also other applications of wireless power transfer for human beings, such as health monitoring and so on; these applications could also be more close to the range of frequency tuou used in this study (an example could be NFC technology 13.56 MHz), some works in literature can be:
Optimizations of source distribution in wireless power transmission for implantable devices
Wireless implantable and biodegradable sensors for postsurgery monitoring: Current status and future perspectives;
Biodegradable and flexible arterial-pulse sensor for the wireless monitoring of blood flow;
- In the introduction “onlological” should be “oncological”;
- Some parameters in table 1 are not clear, as the capacitor values; what do the columns corresponding to + and – mean?
- What do you mean by the expression O.D.?
- What do you mean with MTT? Please explain the acronym;
- Results in Figure 10, for the first column, in the second row picture (after exposition) a lower number of cells seems to be present; in your experimental setup, with the microscope did you always analyze the same spot area?
- In Figure 10, the third column, in the second row , the cells seem to gather in clusters after the EMF exposition;
- Your study is interesting although the frequency of the EMF is a low frequency field (87 kHz); generally the electronic devices that we experiences in our daily lives (cell phones and computers, as you mentioned in the introduction) work at units of GHz, that is much higher frequency; the energy transportation is higher with high frequency, so that being the wavelength smaller the energy interaction with cells is magnified for higher frequency; 87 kHz is radio frequency so it is like the damage to humans due to radio waves, for example to hear the radio stations while we are driving in our cars; the damage for exposition to this type of waves is widely known to be very limited if not completely null, being their wavelength very long with comparison to cell dimensions; so I think yuor study should be much more interesting if conducted with higher frequency close to units of GHz;
- In the discussion you mentioned studies on Wi-Fi electromagnetic waves, that however work at higher frequency so I don’t think that these studies have much in common with your study;
- Also in discussion you mentioned the cell phone applications but, again, they are devices working at different frequency values so I think your study cannot be classified as useful to test if these devices could be harmful to humans;
- Your study should focus on devices working on the same range of frequencies that you used in your experimental setup;
Author Response

(The authors gave the same response as above.)

Reviewer 4 Report
1. Indicate the objectives in the appropriate place.
2. Define a hypothesis.
3. There are some orthographic errors, these are some examples:
- Page 2, paragraph 4, line 1: onlological?
- The scheme of thermos incubator, which displays on the drawning cut in the middle of the incubator (A) ant its front view (B).
- Discussion, line 1: increased?
- Discussion, line 3: smart phones? Smartphones.
Author Response

(The authors gave the same response as above.)

Round 2
Reviewer 2 Report
I am satisfied for the answers and changes in the text provided by authors.
Author Response
Dear reviewer.
We would like to thank you very much for your comments and opinions on our work.
Reviewer 3 Report
The authors answered to the reviewer questions,
the paper was improved with respect to the previous version.
Please in the introduction when you use the expression IM-EMF define what does IM stand for.
Author Response
Dear reviewer.
The IM-EMF within introduction is a mistake, it should be IF-EMF (intermediate frequency electro-magnetic field).
Corrected text has been uploaded as rev_2.